# UPLC-MS/MS Identification and Quantification of Withanolides from Six Parts of the Medicinal Plant *Datura Metel* L.

**DOI:** 10.3390/molecules25061260

**Published:** 2020-03-11

**Authors:** Shi Hui Yang, Yan Liu, Qi Wang, Yan Ping Sun, Wei Guan, Yuan Liu, Bing You Yang, Hai Xue Kuang

**Affiliations:** 1Key Laboratory of Chinese Materia, Heilongjiang University of Chinese Medicine, Ministry of Education, Harbin 150040, China; luckyyangshihui@163.com (S.H.Y.); lifeliuyan@hljucm.net (Y.L.); 18704608056@163.com (Y.P.S.); myguanwei1234@yeah.net (W.G.); flyliuyuan@163.com (Y.L.); 2Department of Medicinal Chemistry and Natural Medicine Chemistry, College of Pharmacy, Harbin Medical University, Harbin 150040, China; mydearmumu@163.com

**Keywords:** *Datura metel* L., UPLC-MS/MS, withanolides, identification, quantification

## Abstract

Withanolides from six parts (flower, leaf, stem, root, seed, and peel) of *Datura metel* L. (*D metel* L.) obtained from ten production areas in China were identified and quantified by UPLC-MS/MS. A total of 85 withanolides were characterized for the first time using the UPLC-Q-TOF-MS/MS system. Additionally, a simultaneous, rapid and accurate measurement method was developed for the determination of 22 bioactive withanolides from ten production areas with the UPLC-Q-TRAP-MS/MS system. The results show the total withanolide content is highest in the leaves (155640.0 ng/g) and lowest in the roots (14839.8 ng/g). Compared with other production areas, the total content of plants from Dujiangyan was the highest at 82013.9 ng/g (value range of ten areas: 82013.9–42278.5 ng/g). The results also show significant differences in the distribution of withanolides in the different plant parts, as well as across different production areas. This is a breakthrough report providing a simultaneous qualitative and quantitative analysis of 22 withanolides in *D. metel* L. It could be the basis for the more rational use of various parts of *D. metel* L., and the expansion of medicinal resources. This work also lays a solid foundation for research on the quality control of *D. metel* L.

## 1. Introduction

Withanolides are highly oxygenated natural products, often occurring in Solanaceae. These C28 steroids with ergostane based skeletons are functionalized at C-22 and C-26, forming six-membered δ-lactone heterocycles. The complex structural features and excellent biological activities of withanolides have attracted the attention of numerous researchers [1,2].

*Datura metel* L. (*D. metel* L.) has abundant withanolides, therefore testing its main pharmacological activities is promising in terms of the potential results. The daturafolisides A, B, F, and I show anti-inflammatory activity. Baimantuoluoside H has significant antiproliferative activity against SGC-7901, Hepg2, and MCF-7 cells [3,4]. Daturafoliside K shows a pronounced effect against NO production [5]. Daturataturin A exhibits cytotoxic activity against SW-620 and MDA-MB-435 cell lines, and potential immunosuppressive activity [6,7].

Though the flower is typically used as the medicinal part of *D. metel* L., it is limited because of the short flowering season and low yield [8]. The factor of different production areas also plays an important role in the diversity and quality of the chemical components of *D. metel* L. Recent research has reported that a large number of withanolides are also found in other parts of *D. metel* L. [9,10]. Given that withanolides possess good physiological activity and are distributed in different parts of the plant, we are very interested in discovering the best parts and the best production areas for *D. metel* L. The non-flowering parts of the plants could potentially be used as other natural sources of withanolides compounds, rather than being treated as waste. However, distribution information regarding the content of different plant parts and species is extremely scarce in relation to the withanolides in *D. metel* L.

Many examples from recent research using qualitative analysis are supported by ultra-performance liquid chromatography tandem time of flight mass spectrometry (UPLC-Q-TOF-MS/MS), including determination of the chemical constituents in herbs [11], antibiotic residue of fruits and vegetables [12,13], and more. Its high sensitivity and fast scanning speed make it a preferred analytical method. To eliminate the effects of the matrix effect in quantitative analysis, the multiple reaction monitoring (MRM) mode in triple quadrupole mass spectrometry (UPLC-QQQ-MS/MS) is frequently employed [14,15]. This method does not require complete separation of the chromatographic peaks of multiple components, as long as the target analysis is non-interfering with each other.

In this experiment, UPLC-Q-TOF-MS/MS and UPLC-QQQ-MS/MS technologies are employed to collect information about the withanolides in six parts (flower, leaf, stem, root, seed, and peel) of *D. metel* L. from ten production areas in China. The objectives of this research are to (1) identify the withanolides from six different parts of *D. metel* L., and (2) explore the content and species differences of 22 withanolides in different parts of *D. metel* L. from 10 different production areas.

This research explores alternative uses of the non-flowering parts of *D. metel* L., and further expands the herb source of this plant. It also provides a basis for the sustainable development and utilization of this resource. Furthermore, a trustworthy, rapid, and accurate quality evaluation system is established in this paper for the control of the medicinal material production areas of *D. metel* L.

## 2. Results and Discussion

### 2.1. Qualitative Analysis of Different Parts of D. metel *L.*

Base peak chromatogram (BPC) graphs of extracts from six parts of *D. metel* L. present obvious withanolides chemical composition differences with the developed UPLC-Q-TOF-MS/MS method (see Electronic Appendix A). A total of 85 withanolides compounds were characterized on the basis of accurate molecular mass, generating molecular ions and fragment ions from UPLC-Q-TOF-MS/MS, and/or by matching this data with our in-house databases and standards. These 85 compounds can be more finely divided into glucosides and non-glucosides, and all of these withanolides have common characteristic fragment ions at *m*/*z* 46, corresponding to formic acid (HCOOH), and *m*/*z* 18 (H_2_O). *m*/*z* 179 [C_12_H_20_O]^−^ is also a distinct characteristic fragment used to demonstrate the presence of withanolides. For example, peak 3 shows a molecular formula of C_30_H_44_O_9_ (m/z 547.29388 [M–H]^−^). Its fragment ions are at *m/z* 501 [M–H^+^-HCOOH]^–^, *m/z* 483 [M–H^+^-HCOOH-H_2_O]^–^, and *m*/*z* 179 [M–H^+^-HCOOH-H_2_O-C_17_H_20_O_5_]^–^, tentatively identified as Baimantuoluoside J according to the in-house databases and standard [16]. Peak 23 shows [M+COOH]^–^ (*m*/*z* 681.34883), demonstrating a molecular formula characterized as C_34_H_52_O_11_. In the MS^2^ spectrum of peak 23, *m*/*z* 470 [M–H^+^-Glc]^–^, 455 [M–H^+^-Glc-H_2_O]^–^, and 179 [M–H^+^-Glc -H_2_O-C_16_H_20_O_4_]^–^ are clearly observed in the negative ion mode. Based on this information, peak 23 is identified as Daturafoliside A [4]. The fragmentation spectrum of peak 26 displays its plus formic acid at *m*/*z* 531.26023 (C_28_H_38_O_7_), *m*/*z* 485 [M–H^+^]^–^, and a series of common fragment ions at 373 (loss of C_6_H_8_O_2_, 112 Da) and 355 (loss of H_2_O, 18 Da). It is tentatively characterized as 5α,12α,27-trihydroxy-(20S,22R)-6α,7α-epoxy-1-oxowitha-2,24-dienolide according to its MS data (Figure 1). The identification data of other compounds are shown in detail in Table 1.

The BPC results show many qualitative differences in the six parts of D. metel L. As we can see from the figures, only eight withanolides (compounds 20, 22, 24, 47, 48, 50, 57, and 80) are present in all parts of *D. metel* L. (Appendix A). The amount of compound 80 (Daturametelin A) is influenced by the production area, ranging from 1671 (Dujiangyan) to 10,538 (Baotou) ng/g (Appendix A). It was first obtained from a methanol extract of the whole plant of *D. metel* L. in 1987. Subsequently, the literature reports that this compound can also be isolated from the seeds of *D. metel* L. [17,18]. The quantitative parts of our experiment also show that the content of compound 80 is generally high in seeds. This result is similar to those reported by several authors for this compound. Compounds 28, 75, and 79 are also distributed in all parts except the seeds. Unfortunately, except for compound 80, because of the scarcity of the standards, none of the remaining compounds (20, 22, 24, 47, 48, 50, 57, 28, 75 and 79) were analyzed for content in this experiment.

### 2.2. Quantitative Analysis of 22 Withanolides by UPLC-Q-TRAP-MS

In previous research, our group isolated a variety of withanolides from different parts of *D. metel* L. From the large amount of isolated withanolides, 22 withanolides were selected as indicators to be tested because of their clear content advantage and better biological activities. The MRM mode of UPLC-Q-TRAP-MS/MS was applied for content determination, and the contents of the analyte were calculated using the constructed calibration curves. Taking the content of the 22 withanolides as the total content, in terms of parts, the total withanolide content of the leaves (*x* = 155640.0 ng/g) was highest, followed by flowers (*x* = 93279.0 ng/g), and the lowest was in the roots (*x* = 14839.8 ng/g) (Figure 2, Appendix A). These results are similar to those reported by several authors for *D. metel* L.; the main antifungal activity found in *D. metel* L. is in the leaves and fruits, whereas the activity in the other parts is very low. The roots are generally less active than the leaves, fruits, and stems [25]. Slight differences may be due to differences in geographical origin and treatment of the parts. In accordance with the literature, this provides us with very convincing data support for the antifungal action of the leaves being mainly related to their withanolide content.

For the total content of withanolides from each production area, the samples were divided into three groups by cluster analysis (Figure 2). S1 (Fuyang), S2 (Haikou), and S7 (Zhaotong) make up one group. S3 (Xingtai), S4 (Dujiangyan), and S5 (Harbin) are another group, and the final group is S6 (Ganzhou), S8 (Baotou), S9 (Baoji), and S10 (Jinhua). Dujiangyan (S4, *x* = 81,789 ng/g), Jinhua (S10, *x* = 67,177 ng/g). and Fuyang (S1, *x* = 41,607 ng/g) are three areas with the top, second, and bottom total withanolide amounts, and they are also evenly distributed in three branch groups. According to Euclidean distances, there are clearly more pronounced differences by region than by plant part (Figure 2). This suggests that Dujiangyan, Jinhua and Ganzhou will be better choices when these withanolides are required for further research (Figure 3). However, because of the influence of plant part differences in each area, although the region difference was visible, there was no significant differences between the areas (*p* < 0.05) according to a one-way analysis of variance (ANOVA) (Figure 3).

Appendix A shows the content distribution of 22 withanolides in various plant parts; the amount of these in different parts is significantly different. For the content of 22 withanolides, the top five accumulations are in the order: compounds 32 (*x* = 91190.2 ng/g), 63 (*x* = 41644.6 ng/g), 64 (*x* = 40943.8 ng/g), 65 (*x* = 33093.8 ng/g), and 69 (*x* = 23999.5 ng/g). Interestingly, the trends of the contents of these five compounds (except 69) in the six parts of *D. metel* L. are similar to the total content (22 kinds of withanolides) in the plant parts (Appendix A). From this, we can speculate that withanolide content differences between parts of *D. metel* L. may have a relationship with the different amounts of these five substances. Among the other compounds, 49, 88, 59, 55, and 13 are the most abundant in the leaves. However, compound 16 has the highest level in flowers, while compounds 46, 78, and 9 are richest in the peels, and 30 and 26 in the stems. Seeds and roots also contain some higher content of the compounds. For the seeds these are 23, 12, and 80, and for the roots these are 36, 72, and 53 (Appendix A).

Meanwhile, although Dujiangyan is the area with the highest total amount, the yield of compound 32 in this area is not abundant, with only 97178.8 ng/g (value range: 43332.8–175155.8 ng/g). Compounds 69 and 63 are the most abundant in Dujiangyan, but both of them are about 13 (value range: 76612.8–5978.5 ng/g) and 25.2 (value range: 108603.1–4312.1 ng/g) times more abundant than in the lowest production area (Haikou and Ganzhou). This finding also applies to lower levels of some compounds, such as 30, 55, and 26 (three of the lowest of the 22 compounds). The value ranges of these three compounds are 102.1 to 203.0 ng/g, 110.4 to 237.9 ng/g, and 574.3 to 1756.0 ng/g. Further, these three ingredients are not all present in all production areas (Appendix A). These results suggest that the geographical area is also an important factor for differences in these compound contents. Some research has confirmed the role of Daturametelin I (64), Daturametelin J (32), and Daturataturin A (65) as a fungicide of charcoal rot fungus, revealing the antiproliferative activity towards the human colorectal carcinoma (HCT-116) cell line of 7α,27-dihydroxy-1-oxowitha-2,5,24-trienolide (69). Daturafoliside O (30), Daturataturin B (55), and 5α,12α,27-trihydroxy-(20S,22R)-6α,7α-epoxy-1-oxowitha-2,24-dienolide (26) also display significant inhibition of nitrite production [5,9]. This gives us some hints that geo-distribution can be an important evaluation indicator when these compounds are required.

On the other hand, we harvested the different plant parts based on the optimal harvesting period, in order to obtain more of the ingredients. The flowers were measured during their flourishing florescence, and the leaves, stems, and roots collected in the vegetative growth stage. The peel and seed were harvested at fruit maturity [8,26]. Additionally, because of the wide geographical differences in the areas of harvest, there were some differences in harvest time (Appendix A). Whether seasonal effects are also an important factor that interferes with withanolide content in different parts of the plants will be the topic of our future research.

## 3. Materials and Methods

### 3.1. Plant Materials and Sample Preparation

Four factors were studied to optimize the extraction procedure, including: extraction methods (heat-reflux, cold-maceration, and ultrasonic extraction), extraction solvents (ethanol:water at 60:40, 70:30, 80:20, and 95:0), extraction times (60 min, 90 min, and 120 min), and extraction repeats (0, 1, and 2). The total withanolide content of the extracts were compared using ultraviolet spectrophotometry. The flower of *D. metel* L. was used to investigate the optimum extraction process. Considering the work efficiency, the optimized extraction procedure was 80% ethanol (ethanol:water at 80:20) for 90 min with heat-reflux, two times (Appendix A).

Six samples were investigated for each production area, and thus a total of 60 batches of *D. metel* L. samples were purchased from ten production areas of China (Appendix A). The herbs were authenticated by Prof. Rui-feng Fan. Voucher specimens were retained in the Chinese Medicine Chemistry Laboratory of Heilongjiang University of Chinese Medicine.

Ten gram samples of each batch from ten production areas were accurately weighed in a mortar with liquid nitrogen. The sample was extracted with 100 mL of ethanol (ethanol:water at 80:20) under heating circumfluence extraction for 90 min, a total of two times. We combined the extracts and then centrifuged at 5000 r/min for 10 min. The supernatant was dried at a low temperature vacuum, then accurately weighed. The sample was dissolved in methanol at 0.5 mg/mL and then stored at –20 °C after sealing.

### 3.2. Chemical, Reagents, Equipment, and Standard Solutions

All the standard substances were isolated and purified by our laboratory [4,5,21,27,28,29] (Figure 4). UPLC analysis determined that the purity of all target compounds was above 98%. HPLC-grade methanol and acetonitrile were supplied from Fisher Chemical (Thermo Fisher Scientific, Waltham, MA, USA). Deionized water was collected from a Milli-Q water purification system (Merck KGaA, Darmstadt, Germany). The rotavapor (N-1300D-W) tandem cold trap (UT-3000A) was from EYELA (EYELA, Tokyo, Japan). An ultrasonic cleaner (B5510E-DTH), high-speed freezing centrifuge (TGL-16M), and electric heater (ZNHW-500) were obtained from Branson (Emerson, MO, USA.), Bioridge (Lu Xiangyi Centrifuge Instrument, Shanghai, China), and Keer (Keer, Shanghai, China) respectively.

Each standard was dispersed in methanol, and the mother liquor, with a concentration of 1 mg/mL, was prepared and sealed in a dark place at 4 °C for one month.

### 3.3. Qualitative UPLC-Q-TOF-MS/MS Analysis

An ACQUITY UPLC^TM^ (Waters Corp., Milford, MA, USA) system in tandem with a Triple TOF^TM^ 5600^+^ mass spectrometer (ABsciex, Framingham, MA, USA.) was acquired for qualitative analysis using a UPLC C_18_ analytical column (2.1 mm × 100 mm, I. D. 1.7 μm, ACQUITY UPLC CSH, Waters Corp., Milford, MA, USA.). The chromatography separation was carried out at an ambient temperature of 25 °C. The mobile phase included water with 0.1% formic acid (A), and acetonitrile with 0.1% formic acid (B). The linear gradient elution was as follows: 0–1 min, 5% B; 1–8 min, 5–30% B; 8–12 min, 30–50% B; 12–25 min, 50–60% B, 25–35 min, 60–95% B; 35–36 min, 95–5% B. The flow rate was set at 0.3 mL/min, with a 5-μL injection volume. The MS/MS parameters were optimized as follows: Scan Type: Positive TOF-MS, TEM: 550, InoSpray Voltage Floating: 4000, DP: 80, CE: 10; Positive Product Ion, TEM: 550, InoSpray Voltage Floating: 4000, DP: 100, CE: 30. Scan Type: Negative TOF-MS, TEM: 550, InoSpray Voltage Floating: 4000, DP: 80, CE: 10; Negative Product Ion, TEM: 550, InoSpray Voltage Floating: 4000, DP: 100, CE: 30.

### 3.4. Method Review

The method validation of the quantitative UPLC-Q-TRAP-MS/MS analysis included linearity, precision (intra- and inter-day), limits of detection (LODs), limits of quantitation (LOQs), repeatability, stability, and recovery. Calibration curves for 22 standards were built by quantitatively diluting the standard solution with methanol in six different multiples. The intra- and inter-day precision were estimated through analysis with six replicates of each standard solution within the same day, and additionally on three consecutive days. The LODs and LOQs were experimentally applied as the minimum concentration, with a detector signal clearly discernable with an S/N of 3 and 10. The flowers were used to evaluate the repeatability, stability, and recovery. Six sample solutions were prepared in parallel for analysis of repeatability. To assess stability, the extract was placed at room temperature for 0, 2, 4, 6, 12, 24, and 48 h for analysis. Recovery tests were performed by adding approximately 50%, 100%, and 150% of the known chemical markers to the sample, accomplishing six independent spiked analyses. The formula of recoveries is: Recovery (%) = (Measured value of the spiked sample – Measured value of the sample)/(Amount spiked) × 100%.

All analytes with linearity were better than *r*^2^ = 0.994, indicating a good linear relationship between the analyte concentrations and their peak area within the test ranges. The RSDs of intra- and inter-day variations were within 0.37–4.03% and 0.83–4.11% for the 22 analytes, respectively. The repeatability was less than 4.98% (RSDs), and stability lower than 3.49% (RSDs) (Table 2). These results distinctly demonstrate that determination of the 22 reference compounds in different parts of *D. metel* L. can undoubtedly be achieved with the developed quantitative UPLC-Q-TRAP-MS/MS method, with good linearity, precision, stability, sensitivity, repeatability, and accuracy.

### 3.5. Quantitation for UPLC-Q-TRAP-MS/MS Analysis

The target analytes were found using the MRM method (negative ionization mode). Using the direct infusion mode, [M–H]^–^ was selected as the precursor ion of all withanolides to determine the cone energy (CE) and declustering potential (DP) (Appendix A). At the same time, different elution conditions were further investigated to achieve good resolution of the 22 withanolides. Ultimately, the best elution system was acetonitrile (0.1% formic acid) and water (0.1% formic acid) at 54:46 (*v*/*v*).

The quantitation analysis was done using an ACQUITY UPLC^TM^ (Waters Corp., Milford, MA, U.S.A.) system tandem 4000 Q-TRAP mass spectrometer (ABsciex, Framingham, MA, U.S.A.) with the same analytical column. The column temperature was 25 °C. Although the mobile phase was the same as in the qualitative analysis, isocratic elution was performed as follows: 0–6 min, 54% B, with 3 μL injection volume. Detection was performed in the negative electrospray ionization mode (ESI^−^) in the MRM. Appendix A shows the mass spectrometer conditions.

### 3.6. Analysis of UPLC-MS/MS Data

Analyst 1.6.4 (ABsciex, Framingham, MA, U.S.A.) software controlled the UPLC-Q-TOF-MS/MS and UPLC-Q-TRAP-MS/MS systems. The qualitative data were processed using Peakview 2.1 (ABsciex, Framingham, MA, USA.) software. Group differences were determined using a one-way analysis of variance (ANOVA), performed in GraphPad Prism 7 (GraphPad, San Diego, CA, USA.), and *p* < 0.05 was considered statistically significant. Hierarchical clustering and the correlation heat-map were constructed using the R language package of the OmicsShare platform (https://www.omicshare.com/tools/).

## 4. Conclusions

In this study, all parts (flower, leaf, peel, stem, seed and root) of the herb *D. metel* L. were subjected to a detailed withanolide qualification using a newly developed UPLC-Q-TOF-MS/MS method. A total of 85 withanolides were characterized for the first time, and a simultaneous, rapid, and accurate measurement method was developed for the determination of 22 bioactive withanolides from ten production areas with the UPLC-Q-TRAP-MS/MS system. The results show the significant differences in the distribution of withanolides in different parts of the plant, as well in plants from different production areas. The total content of withanolides is the highest in the leaves, followed by the flowers and peel. Overall, in terms of the withanolide content of each production area, Dujiangyan and Jinhua stand out among the ten production areas. This is a breakthrough report featuring a simultaneous qualitative and quantitative analysis of 22 withanolides in *D. metel* L. It provides a basis for the more rational use of the various parts of *D. metel* L. and the expansion of medicinal resources. This work also lays a solid foundation for future research on the quality control of *D. metel* L.

## Figures and Tables

**Figure 1 molecules-25-01260-f001:**
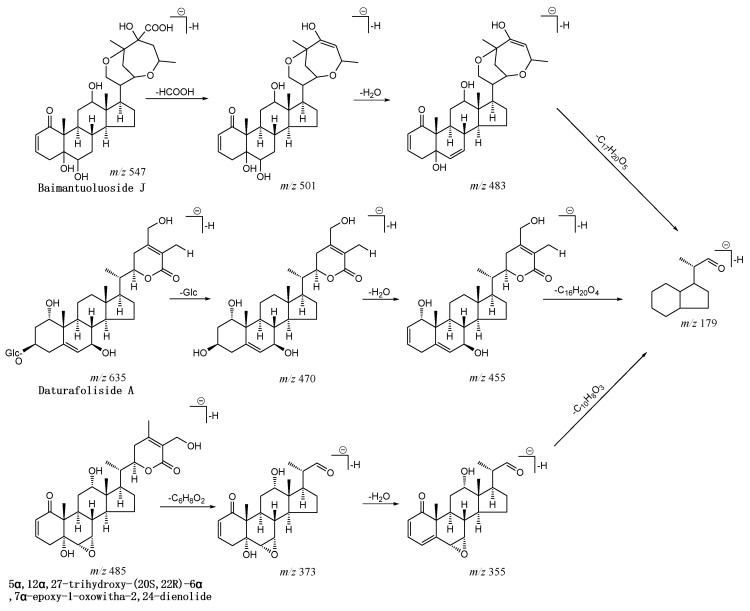
The proposed fragmentation pathways of three example withanolides from *Datura metel* L. (*D. metel* L.).

**Figure 2 molecules-25-01260-f002:**
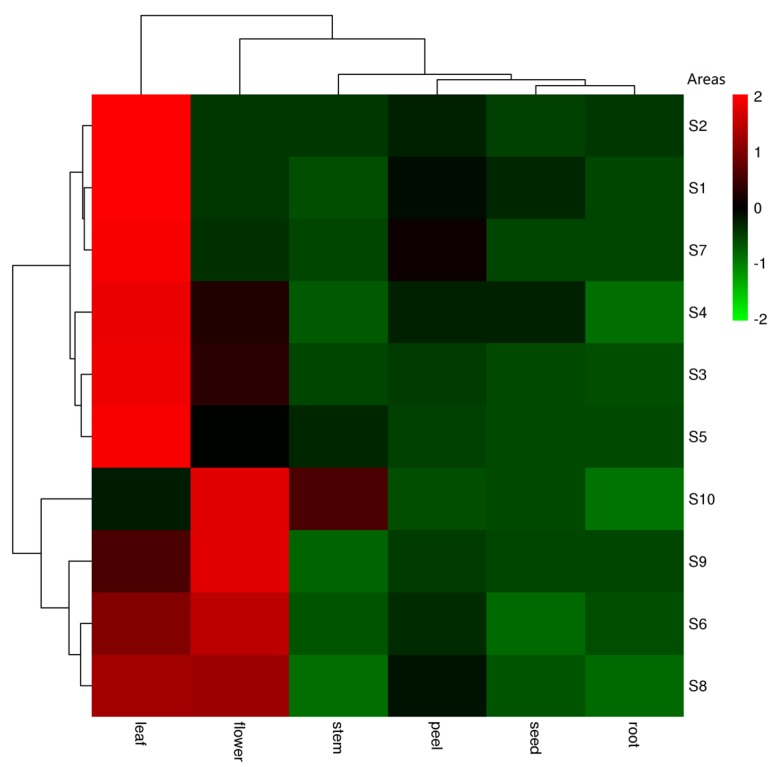
The heat-map of total withanolide content in each part of *D. metel* L.

**Figure 3 molecules-25-01260-f003:**
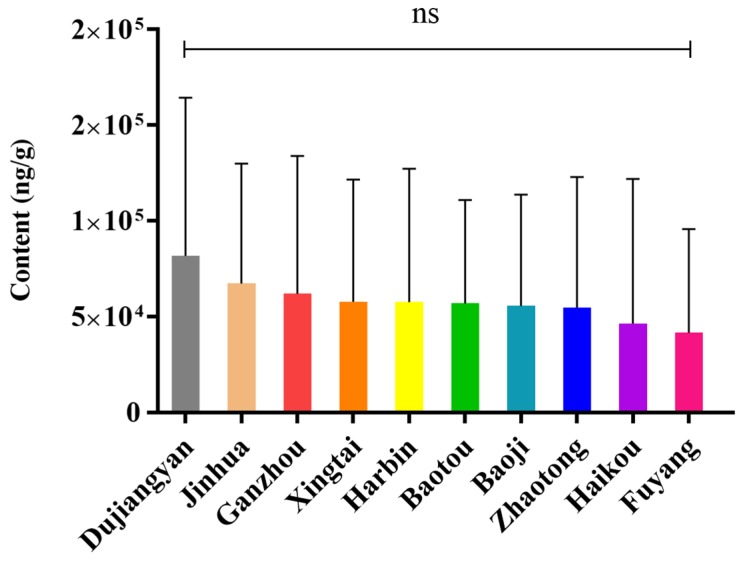
Total withanolide content in each production area of *D. metel* L. (ns no significance).

**Figure 4 molecules-25-01260-f004:**
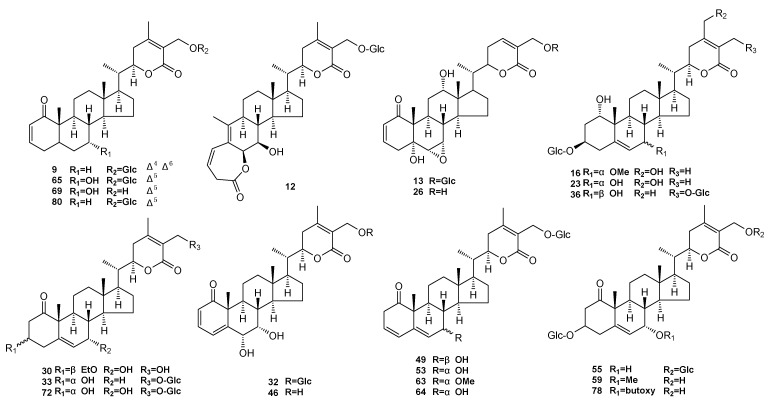
Chemical structures of the 22 withanolides.

**Table 1 molecules-25-01260-t001:** Withanolide identification data of *Datura metel* L. (*D. metel* L.) by ultra-performance liquid chromatography tandem time of flight mass spectrometry (UPLC-Q-TOF-MS/MS).

Peak No.	t_R_ (min)	*m*/*z* [M ± H]^+/–^	Formula	Compound [Ref.]	Fragment ions	Adduct	Error (ppm)
1	3.6	547.7	C_29_H_40_O_8_S	Daturametelin E [19]	565, 519, 501	+OH^–^	4.3
2	4.3	469.6	C_29_H_42_O_5_	1α,3β,27-trihydroxy-(20S,22R)-3-methoxywitha-4,6,24-trienolide	487, 397	+OH^–^	1.9
3	4.4	547.6	C_30_H_44_O_9_	Baimantuoluoside J [16]	547, 501, 483	–H^+^	4.8
4	4.7	469.6	C_28_H_38_O_6_	5α,27-dihydroxy-6α,7α-epoxy-1-oxowitha-2,24-dienolide	487, 441, 397	+OH^–^	1.6
5	4.7	505.6	C_28_H_40_O_8_	5α,6β,12β,21,27-pentahydroxy-(22R)-1-oxowitha-2,24-dienolide [20]	543, 497, 479	+K^+^	1.6
6	5.0	797.9	C_40_H_62_O_16_	Daturafoliside V [21]	833, 773	+Cl^–^	0.2
7	5.0	799.9	C_40_H_62_O_16_	Daturafoliside W [21]	837, 791	+K^+^	0.2
8	5.1	665.7	C_34_H_50_O_13_	Baimantuoluoside F [20]	701, 655	+Cl^–^	–0.1
9	5.1	599.7	C_34_H_46_O_9_	Baimantuoluoside H [20]	621, 575	+Na^+^	–0.1
10	5.1	635.7	C_34_H_52_O_11_	1α,3β,7α,12α,27-pentahydroxy-(20S,22R)-witha-5,24-dienolide-3-*O*-β-d-glucopyranoside [5]	635, 545	–H^+^	4.7
11	5.2	631.7	C_34_H_48_O_11_	5α,27-dihydroxy-(20S,22R)-6α,7α-epoxy-1-oxowitha-2,24-dienolide-27-*O*-β-d-glucopyranosy	649, 631	+OH^–^	1.3
12	5.2	631.7	C_34_H_48_O_11_	Daturafoliside K [5]	649, 603	+OH^–^	1.3
13	5.2	647.7	C_34_H_48_O_12_	Baimantuoluoside B [20]	647, 601	–H^+^	–3
14	5.6	633.7	C_34_H_50_O_11_	Baimantuoluoside G [20]	633, 587, 543	–H^+^	2.3
15	5.6	615.7	C_34_H_48_O_10_	Daturametelin B	633, 543, 179	+OH^–^	3.2
16	5.7	649.8	C_35_H_54_O_11_	Daturafoliside B [4]	649, 631, 179	–H^+^	–3.4
17	5.8	635.7	C_34_H_52_O_11_	1α,3β,7α,28-tetrahydroxy-(20S,22R)-witha-5,24-dienolide-3-*O*-β-d-glucopyranoside	653, 607, 428	+OH^–^	–4.1
18	5.8	651.7	C_34_H_52_O_12_	Daturameteloside J	697, 679, 633	+COOH^–^	–3.4
19	6.3	533.7	C_30_H_44_O_8_	Baimantuoluoline I [16]	555, 509	+Na^+^	2.4
20	6.5	631.7	C_34_H_48_O_11_	1,10-seco-6β,7α,27-trihydroxy-(20S,22R)-witha-3,5,24-trienolide-1-oicacid-ε-lactone-3-*O*-β-d-glucopyranoside	667, 621	+Cl^–^	–2.2
21	6.5	781.9	C_40_H_62_O_15_	Daturafoliside U [5]	781, 735	–H^+^	–2.2
22	6.6	631.7	C_34_H_48_O_11_	Daturafoliside M [5]	677, 631, 525	+COOH^–^	–0.1
23	6.7	635.7	C_34_H_52_O_11_	Daturafoliside A [4]	681, 635, 179	+COOH^–^	0.3
24	6.7	635.7	C_34_H_52_O_11_	Daturafoliside T [5]	681, 663	+COOH^–^	–2.3
25	6.7	485.6	C_28_H_38_O_7_	Baimantuoluoline A [20]	531, 471	+COOH^–^	4.4
26	6.7	485.6	C_28_H_38_O_7_	5α,12α,27-trihydroxy-(20S,22R)-6α,7α-epoxy-1-oxowitha-2,24-dienolide	531, 485, 355	+COOH^–^	1.5
27	6.7	635.7	C_34_H_52_O_11_	Daturameteloside I [22]	681, 635, 617	+COOH^–^	–0.4
28	6.8	690.8	C_38_H_59_O_11_	Daturafoliside H [4]	690, 644	–H^+^	–0.8
29	6.8	689.8	C_38_H_58_O_11_	3β,7α,27-trihydroxy-3-Obutyl-(20S,22R)-1-oxowitha-5,24-dienolide-27-*O*-β-d-glucopyranoside	689, 643, 179	–H^+^	4.2
30	6.8	689.8	C_38_H_58_O_11_	Daturafoliside O [5]	689, 643,599	–H^+^	4.2
31	7.7	649.7	C_34_H_50_O_12_	Baimantuoluoside D [20]	649, 631, 585	–H^+^	2.2
32	7.7	631.7	C_34_H_48_O_11_	Daturametelin J [21]	649, 631, 587	+OH^–^	3
33	8.5	617.7	C_34_H_50_O_10_	Daturafoliside Q [5]	635, 545	+OH^–^	–1.1
34	8.8	635.7	C_34_H_52_O_11_	1α,3β,7β,27-tetrahydroxy-(20S,22R)-witha-5,24-dienolide-3-*O*-β-d-glucopyranoside	635, 589, 179	–H^+^	–2.4
35	8.8	633.7	C_34_H_50_O_11_	3β,7α,27-trihydroxy-(20S,22R)-1-oxowitha-5,24-dienolide-27-*O*-β-d-glucopyranoside	633, 543	–H^+^	–3.3
36	8.8	615.7	C_34_H_48_O_10_	Daturafoliside D [4]	633, 587	+OH^–^	–2.4
37	8.8	633.7	C_34_H_50_O_11_	Daturafoliside N [5]	633, 615	–H^+^	–1.3
38	9.2	473.6	C_28_H_42_O_6_	Baimantuluodine K [23]	519, 473, 455	+COOH^–^	–1.9
39	9.2	473.6	C_28_H_42_O_6_	Baimantuoluoline K	519, 501	+COOH^–^	–0.1
40	10.0	469.6	C_28_H_38_O_6_	Baimantuoluoline G [20]	487, 441	+OH^–^	–0.8
41	10.0	487.6	C_28_H_40_O_7_	Daturameteline H	487, 441	–H^+^	–1.9
42	10.6	469.6	C_28_H_38_O_6_	5α,6β-dihydroxy-21,24-epoxy-1-oxowitha- 2,25(27)-dienolide	469, 423	–H^+^	–3.4
43	10.6	517.6	C_29_H_42_O_8_	Baimantuoluoline C [20]	517, 471	–H^+^	–2.2
44	10.6	647.7	C_34_H_48_O_12_	5α,12α,27-trihydroxy-(20S,22R)-6α,7α-epoxy-1-oxowitha-2,24-dienolide-27-*O*-β-d-glucopyranosy	647, 601	–H^+^	0.4
45	10.6	647.7	C_34_H_48_O_12_	Baimantuoluoside A [20]	647, 601	–H^+^	0.4
46	10.9	469.6	C_28_H_38_O_6_	Daturafoliside S [5]	469, 423	–H^+^	–3.4
47	11.2	647.7	C_35_H_52_O_11_	3β,7α,27-trihydroxy-3-methoxy-(20S,22R)-1-oxowitha-5,24-dienolide-27-*O*-β-d-glucopyranoside	693, 647, 629	+COOH^–^	–0.3
48	11.3	647.7	C_35_H_52_O_11_	Daturafoliside G [4]	665, 619	+OH^–^	–2.6
49	11.3	615.7	C_34_H_48_O_10_	Daturafoliside I [4]	615, 597, 551	–H^+^	–2.6
50	11.4	619.7	C_34_H_52_O_10_	Daturametelin N	619, 573	–H^+^	–2.1
51	11.4	473.6	C_28_H_40_O_6_	Acnistoferin	473, 427	+H^+^	–0.3
52	11.4	547.6	C_30_H_44_O_9_	Baimantuoluoline J [20]	547, 501, 483	–H^+^	–3.1
53	12.1	617.7	C_34_H_48_O_10_	7α,27-dihydroxy-(20S,22R)-1-oxowitha-2,5,24-trienolide-27-*O*-β-d-glucopyranosy	634, 588	+NH_4_^+^	2.8
54	12.1	783.8	C_42_H_56_O_14_	Daturameteline G-Ac [19]	829, 783, 739	+COOH^–^	3.7
55	12.2	635.7	C_34_H_52_O_11_	Daturataturin B [21]	635, 589, 571	–H^+^	–1.6
56	12.2	691.8	C_38_H_60_O_11_	1α,3β,7β,27-tetrahydroxy-(20S,22R)-7-butoxywitha-5,24-dienolide-3-*O*-β-d-glucopyranoside	727, 681, 637	+Cl^–^	–3.3
57	12.2	691.8	C_38_H_60_O_11_	Daturafoliside E [4]	727, 681, 663	+Cl^–^	–3.3
58	12.3	649.8	C_35_H_54_O_11_	1α,3β,7α,2-tetrahydroxy-7-methoxy-(20S,22R)-witha-5,24-dienolide-3-*O*-β-d-glucopyranoside	649, 603, 585	–H^+^	–3.2
59	12.3	649.8	C_35_H_54_O_11_	daturafoliside Y [23]	649, 631, 587	–H^+^	–4.4
60	12.6	757.8	C_40_H_54_O_14_	(20S,22R)-witha-1,3,5,6,8,24-hexaenolide-3,27-*O*-β-d-diglucopyranoside	757, 711, 667	–H^+^	–3.7
61	12.6	469.6	C_29_H_40_O_5_	Daturametelin C [19]	491, 445, 401	+Na^+^	–3.7
62	12.6	505.6	C_28_H_42_O_8_	Baimantuoluoline D [20]	523, 477	+OH^–^	–2.4
63	13.0	615.7	C_34_H_48_O_10_	7α,27-dihydroxy-(20S,22R)-7-methoxy-1-oxowitha-3,5,24-trienolide-27-*O*-β-d-glucopyranosy	615, 569, 551	–H^+^	–0.8
64	13.0	615.7	C_34_H_48_O_10_	Daturametelin I [23]	615, 597, 551	–H^+^	–4.7
65	13.0	615.7	C_34_H_48_O_10_	Daturataturin A [21]	651, 605	+Cl^–^	–4.3
66	13.2	631.7	C_34_H_48_O_11_	Baimantuoluoside C [20]	631, 585	–H^+^	–1.2
67	13.2	635.7	C_34_H_50_O_11_	Daturafoliside P [5]	652, 635, 589	+NH_4_^+^	–1.2
68	13.4	631.7	C_34_H_48_O_11_	6α,7α,27-trihydroxy-(20S,22R)-1-oxowitha-2,4,24-trienolide-27-*O*-β-d-glucopyranosy	667, 621, 603	+Cl^–^	–1.3
69	13.4	455.6	C_28_H_38_O_5_	7α,27-dihydroxy-1-oxowitha-2,5,24-trienolide	493, 455, 409	+K^+^	–1.3
70	14.0	629.7	C_35_H_50_O_10_	(22R)-27-hydroxy-7α-methoxy-1-oxowitha-3,5,24-trienolide	647, 601, 583	+OH^–^	–3.9
71	15.2	471.6	C_28_H_40_O_6_	Hyoscyamilactol [24]	471, 425	–H^+^	–1.2
72	15.3	633.7	C_34_H_50_O_11_	Daturafoliside F [4]	633, 587	–H^+^	–2.9
73	15.3	629.7	C_35_H_50_O_10_	Daturafoliside R [5]	629, 569	–H^+^	–2.9
74	16.2	649.7	C_34_H_50_O_12_	Baimantuoluoside E [20]	649, 603	–H^+^	–4.1
75	16.2	597.7	C_34_H_46_O_9_	Daturametelin K [22]	597, 551, 533	–H^+^	–4.1
76	16.3	629.7	C_35_H_50_O_10_	Daturametelins L [22]	629, 585, 539	–H^+^	–3.2
77	17.9	637.8	C_34_H_52_O_11_	Daturafoliside C [4]	637, 591	+H^+^	–4.5
78	17.9	599.7	C_34_H_48_O_9_	Daturafoliside X [23]	599, 553, 535	–H^+^	–4.5
79	19.2	473.6	C_28_H_40_O_6_	5α,6β,21,27-tetrahydroxy-1-oxowitha-2,24-dienolide	473, 427, 409	+H^+^	–2.5
80	19.2	599.7	C_34_H_48_O_9_	Daturametelin A [4]	599, 553,509	–H^+^	–2.5
81	20.8	469.6	C_29_H_40_O_5_	Daturametelin D [19]	469, 423, 405	+H^+^	–3.3
82	21.1	533.6	C_28_H_38_O_8_S	Daturametelin F [19]	533, 473	–H^+^	–0.7
83	25.5	503.6	C_28_H_40_O_8_	Baimantuoluoline E [20]	521, 503	+OH^–^	–4.7
84	30.2	473.6	C_28_H_42_O_6_	Daturametelin M [22]	473, 427	–H^+^	–4.9
85	34.1	619.7	C_33_H_46_O_11_	Daturafoliside L [5]	619, 601, 555	+H^+^	3.7

**Table 2 molecules-25-01260-t002:** Calibration curves and method validation data of the 22 bioactive withanolides by UPLC-Q-TRAP-MS/MS.

Peak No.	Compound	Calibration Curve	r^2^	Linear Range ^a^	LOD ^a^	LOQ ^a^	Precision ^b^		Stability ^c^	Repeatability ^b^	Recovery ^d^		
							Intraday	Interday			50%	100%	150%
9	Baimantuoluoside H	y = 32.588x – 814.6	0.9928	100–3200	33	100	3.6	0.9	2.6	1.6	49.8	101.4	152.4
12	Daturafoliside K	y = 1.6259x – 1462	0.9952	500–16000	165	500	2.9	2.1	3.2	1.3	50.9	101.2	151.4
13	Baimantuoluoside B	y = 2.6717x + 316.35	0.9957	100–3200	33	100	1.3	2.5	3.5	2.2	51.1	101.3	150.7
16	Daturafoliside B	y = 110.99x – 10552	0.9979	150–4800	50	150	3.5	3.0	1.1	1.9	50.2	103.2	151.5
23	Daturafoliside A	y = 1.0974x + 111.17	0.9962	100–3200	33	100	2.3	2.1	2.3	3.7	49.9	100.4	150.4
26	5α,12α,27-trihydroxy-(20S,22R)-6α,7α-epoxy-1- oxowitha-2,24-dienolide	y = 1.5217x – 243.83	0.9967	200–1000	67	200	2.0	3.9	0.8	4.1	49.0	100.5	149.7
30	Daturafoliside O	y = 657.86x – 57031	0.9973	100–3200	33	100	2.6	2.2	2.5	4.4	50.1	99.5	149.7
32	Daturametelin J	y = 2.4051x – 2739.4	0.9955	700–22400	234	700	3.3	3.1	1.6	3.5	50.3	99.7	150.4
33	Daturafoliside Q	y = 15.05x + 421.69	0.9949	150–4800	50	150	2.5	1.6	1.8	5.0	50.5	100.3	150.5
36	Daturafoliside D	y = 0.3519x + 65.662	0.9982	100–3200	33	100	2.8	1.6	0.9	3.3	49.0	100.4	150.1
46	Daturafoliside S	y = 0.3792x – 565.79	0.9935	2000–10000	667	2000	1.9	1.4	1.6	1.5	48.2	99.9	151.3
49	Daturafoliside I	y = 0.4324x + 98.775	0.994	100–3200	33	100	0.7	1.2	1.0	4.7	48.9	99.9	150.0
53	7α,27-dihydroxy-(20S,22R)-1-oxowitha-2,5,24- trienolide-27-*O*-β-d-glucopyranosy	y = 0.1799x + 57.304	0.9959	200–6400	67	200	4.0	2.8	1.3	2.8	50.4	101.6	149.5
55	Daturataturin B	y = 139.02x – 6747.7	0.9951	100–3200	33	100	1.5	2.0	1.5	4.0	49.8	100.6	151.2
59	Daturafoliside Y	y = 20.538x – 8567.2	0.9974	600–19200	2000	6000	1.7	1.1	2.2	1.9	49.6	101.7	149.9
63	7α,27-dihydroxy-(20S,22R)-7-methoxy-1-oxo- witha-3,5,24-trienolide-27-*O*-β-d-glucopyranosy	y = 28.565x – 15872	0.9982	400–12800	134	400	0.9	1.6	1.5	3.1	50.7	100.3	150.2
64	Daturametelin I	y = 192.09x – 106108	0.9946	1400–44800	467	1400	3.7	2.7	3.4	2.3	50.3	101.3	150.0
65	Daturataturin A	y = 193.45x + 12937	0.9946	1400–44800	467	1400	3.4	2.5	2.4	2.7	52.0	98.4	150.5
69	7α,27-dihydroxy-1-oxowitha-2,5,24-trienolide	y = 18.916x – 8803	0.9955	1000–32000	334	1000	1.7	0.8	2.5	1.4	50.9	99.8	151.2
72	Daturafoliside F	y = 5.8692x – 157.78	0.9972	100–3200	33	100	1.9	2.4	2.3	4.5	49.6	100.5	151.4
78	Daturafoliside X	y = 0.2666x + 72.031	0.9953	400–12800	133	400	3.7	4.1	3.3	4.0	49.6	102.6	151.6
80	Daturametelin A	y = 3.0847x + 775.02	0.9974	300–9600	100	300	0.4	1.4	0.8	1.2	50.3	99.9	150.9

^a^ ng/mL; ^b^
*n* = 6, RSD, %; ^c^ 48 h, RSD, %; ^d^
*n* = 6, %.

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
