# Peer review of "UPLC-MS/MS Identification and Quantification of Withanolides from Six Parts of the Medicinal Plant Datura Metel L."

_molecules, 2020, doi:10.3390/molecules25061260_

Round 1
Reviewer 1 Report
General comments.
In the paper “Ultra-High Performance Liquid Chromatography Tandem Mass Spectrometry Identification and Quantification of The Withanolides from Six Positions Derived from Datura Metel L.” authors reported the characterization of Datura metel L by UPLC-Q-TOF-MS/MS.
Author reported the characterization of 85 withanolides and found significant differences in the distribution of withanolides in different positions as well as the producing
Paper is good write and supported by results. I suggest minor revisions.
Specific comments.
Page 2 line
“Many examples from recent literatures on qualitative analysis were supported by the ultra-performance liquid chromatography tandem time of flight mass spectrometry (UPLC-Q-TOF-MS/MS), including determination of chemical constituents in herb [9], 56 antibiotic-residue of fruits and vegetables [10].”
Please, add other references such as
Determination of estrogenic endocrine disruptors in water at sub-ng L− 1 levels in compliance with Decision 2015/495/EU using offline-online solid phase extraction concentration coupled with high performance liquid chromatography-tandem mass spectrometry. Microchemical Journal, 147, 1186-1191.
And in the sentence
Its high sensitivity and fast scanning speed were the reasons why it was selected as the analytical method. To eliminate the effects of the matrix effect in quantitative analysis, the multiple reaction monitoring (MRM) mode in triple quadrupole mass spectrometry (UPLC-QQQ-MS/MS) was frequently employed for quantitative analysis [11].
Please, add other references such as
Online solid-phase extraction LC-MS/MS: a rapid and valid method for the determination of perfluorinated compounds at sub ng· L− 1 Level in natural water. Journal of Chemistry, 2018.
Reviewer 2 Report
The authors must modify certain aspects in their manuscript according to the following suggestions, in order to improve its scientific soundness and novelty.
General. Sometimes, the manuscript's syntax and spelling are poor (e.g. past tense is misused throughout the text). It needs to be reviewed again by an English-speaking native person or by a certified translation agency. Avoid as much as possible self-citations. Certain references are not formatted according to Molecules’ instruction for authors neither is the author’s contribution statement (https://www.mdpi.com/journal/molecules/instructions).
Title. Too long; suggestion: UPLC-Q-TOF-MS/MS identification of withanolides from Devil’s trumpet (Datura metel L).
Abstract. Very long. It is suggested to reconstruct this section highlighting the differential effects of geo-distribution and at each part of the plant on the natural occurrence of withanolides; include the most relevant data in a more quantitative way, indicating the significant differences. Be aware that the abstract should be up to 200 words max.
Figures. The following changes are strongly suggested:
All chromatograms included as Figure 1 should be included as supplementary material (or eliminated) since each the same information is mentioned in Table 1. What's the purpose of including Figure 2, particularly if it is not discussed within the plant's physiology context (e.g. where does it occur? which are the most plausible enzymatic mechanisms?). Figure 3, the total content of withanolides (based in the referred 22 structures in Table 2 and Figure 5 and lines 173-180) is OK, but wouldn’t be preferable an image (e.g. a heat-map) showing how are these molecules distributed in each plant part?. Figure 4, it is OK but please arrange bars from highest (Dujiangyan) to lowest (Fuyang) Figure 5, please annotate the correspondence of all 22 withanolides (lines 173-180) with that coding system used in Table 1.
Tables. The following changes are strongly suggested:
Table 1: Include values with just one decimal. Arrange as follows: Peak number-tR (min)- m/z [M-H]+ or –Formula-Compound (ref)-Fragments (just values)-Adduct-Error. Table 2: which withanolide structure correspond to markers 1-22?, what was the basis for their selection?, are they evenly (though in different concentration) distributed in each plant part?.
Results and discussion. This section is the cornerstone, and the study's uniqueness relies on it. Please review and modify the following:
It is the opinion of this reviewer that all information regarding method optimization (sections 2.1, 2.3, and 2.4 and related Tables or Figures) should be located in section 3 (materials and methods) since they do not contribute to the overall discussion, besides being described as incidental data. (2) Authors do not explore statistical methods in order to weigh the effect of geographical origin and plant part on the natural occurrence of specific/different molecules. For example, what do you mean with “obvious differences” in chemical composition from the different position (Line 84)”?, at least compounds 48 and 50 (all), 79 (all but seeds) and 53 (all but peel) are evenly distributed. You stated having identified 85 withanolides (Table 1), yet peaks 86 and 90 were identified in seeds (OF-ESC+) but were not included in Table 1 nor discussed properly. (4) Results are just described but not discussed properly. Authors should take a look to other supporting reports including yours (DOI): 1002/hlca.200790159 , 10.1007/s12272-001-1274-6 , https://pdfs.semanticscholar.org/b4b8/1ab337051ae897ef9136c557eb24d5c2d350.pdf.
Methods. Please change the following:
Section 3.1. How many plant samples (in fact they should be termed as accessions) correspond to each geographical location?. All equipment (and manufacturers) should be properly annotated (Heating circumfluencer, centrifuge, rotary evaporator, etc.). Section 3.2. Include structure annotations (lines 173-180) as figure footnote and eliminate from body text.
Conclusions
You stated, “This was the first qualitative and quantitative analysis of the withanolides in metel L.” – It is not (references 2, 3, 12, 14, 15, 17, 19, 20, 21).
Author Response
Point 1: Sometimes, the manuscript's syntax and spelling are poor (e.g. past tense is misused throughout the text). It needs to be reviewed again by an English-speaking native person or by a certified translation agency.
Response 1: Done. Thanks for your correction, our manuscript has been reviewed by a native English speaker to improve the gramma/wording style. (marked red in manuscript)
Point 2: Avoid as much as possible self-citations. Certain references are not formatted according to Molecules’ instruction for authors neither is the author’s contribution statement (https://www.mdpi.com/journal/ molecules/instructions).
Response 2: Done. Thank you very much for your valuable comments, and we have added the related reference and formatted according to Molecules’ instruction in our manuscript (marked red in references 3, 6, 12, 14, 17, 18, 25, 26 and 27). We have also formatted certain references and added author contributions as well as conflicts of interest after the conclusion part of the manuscript. (marked red in lines 256-258 and 265-266).
Point 3: Title. Too long; suggestion: UPLC-Q-TOF-MS/MS identification of withanolides from Devil’s trumpet (Datura metel L).
Response 3: Done. Thank you for your professional insights. Combined your comments with other reviewers’, the title is revised as: UPLC-MS/MS Identification and Quantification of Withanolides from Six Parts of the Medicinal Plant Datura Metel L. (marked red in lines 2-4)
Point 4: Abstract. Very long. It is suggested to reconstruct this section highlighting the differential effects of geo-distribution and at each part of the plant on the natural occurrence of withanolides; include the most relevant data in a more quantitative way, indicating the significant differences. Be aware that the abstract should be up to 200 words max.
Response 4: Done. Thank you for your professional insights, and we have revised and condensed Abstract in our manuscript. (marked red in lines 13-26)
Point 5: Figures. The following changes are strongly suggested:
(1) All chromatograms included as Figure 1 should be included as supplementary material (or eliminated) since each the same information is mentioned in Table 1.
(2) What's the purpose of including Figure 2, particularly if it is not discussed within the plant's physiology context (e.g. where does it occur? which are the most plausible enzymatic mechanisms?).
(3) Figure 3, the total content of withanolides (based in the referred 22 structures in Table 2 and Figure 5 and lines 173-180) is OK, but wouldn’t be preferable an image (e.g. a heat-map) showing how are these molecules distributed in each plant part?
(4) Figure 4, it is OK but please arrange bars from highest (Dujiangyan) to lowest (Fuyang)
(5) Figure 5, please annotate the correspondence of all 22 withanolides (lines 173-180) with that coding system used in Table 1.
Response 5: Done. Thank you for your professional insights.
(1) we have transferred Figure 1 to our supplementary material. (Figure S1)
(2) The structures of withanolides are identified according to its MS data and its racking law. In section 2.1, we have identified three withanolides compounds basing on their MS data. The presence of Figure 2 is the visual proposed fragmentation pathway of these three withanolides. Certainly, we redrawn Figure 2 to help readers digest and understand more clearly.
(3) We have changed Figure 3 into a heat-map in our manuscript. (Figure 2)
(4) We have adjusted the order of Figure 4 in our manuscript. (Figure 3)
(5) We have adjusted the numbers of 22 withanolides with that coding system used in Table 1. in our manuscript. (Figure 4)
Point 6: Tables. The following changes are strongly suggested:
(1) Table 1: Include values with just one decimal. Arrange as follows: Peak number-tR (min)- m/z [M-H]+ or –Formula-Compound (ref)-Fragments (just values)-Adduct-Error.
(2) Table 2: which withanolide structure correspond to markers 1-22? what was the basis for their selection?
(3) Are they evenly (though in different concentration) distributed in each plant part?
Response 6: Done. Thank you for your professional insights.
(1) We have adjusted the order of Table 1 and made the values of tR (min) and [M-H]+/– with one decimal in our manuscript. (marked red in Table 1)
(2) we have adjusted the markers and made their serial number same as the coding system of Table 1. (marked red in Table 2) We have also adjusted the numbers of all Tables and Figures with that coding system used in Table 1. in our electronic supplementary material. (marked red in Table S1, Table S4, Figure S2 and Figure S4)
The mainly reason of makers selection is that they are the main contributors to the biological activity of D. metel L.. For example, Baimantuoluoside H reflects significant antiproliferative activity against SGC-7901, Hepg2 and MCF-7 cells. Daturafolisides A, B, F and I perform nice anti-inflammatory activity [1,2]. Some literatures confirm the role of Daturametelin I, Daturametelin J and Daturataturin A as a fungicide of charcoal rot fungus, revealing the antiproliferative activity towards the human colorectal carcinoma (HCT‐116) cell line ability of 7α,27-dihydroxy-1-oxowitha-2,5,24-trienolide. Daturafoliside K shows pronounced effect against NO production. Daturafoliside O, Daturataturin B and 5α,12α,27-trihydroxy-(20S,22R)-6α,7α–epoxy-1-oxowitha-2,24-dienolide also display significant inhibition of nitrite production [3,4]. Daturataturin A exhibits cytotoxic activity against SW-620 and MDA-MB-435 cell lines and potential immunosuppressive activity [5,6]. At the same time, our group have isolated a variety of withanolides from different parts of D. metel L. in previous research. From the large amount of isolated withanolides, these compounds also perform clear content advantage. To sum up, these are why we select their as indicators for quantitative analysis.
We have added the corresponding explanation in introduction and 2.2 part of our manuscript. (marked red in lines 35-39 and 146-153)
(3) We have conducted a statistical study on the distribution of each marker (1-22) in six parts, and the results shows that except for 2, 7, 8, 16 and 19, other markers are evenly distributed in all parts. (Figure S2) But the markers concentration is usually obvious different based on those different geographical origins. We have further added the relevant discussion in our manuscript. (marked red in lines 127-146)
Point 7: Results and discussion. This section is the cornerstone, and the study's uniqueness relies on it. Please review and modify the following:
(1) It is the opinion of this reviewer that all information regarding method optimization (sections 2.1, 2.3, and 2.4 and related Tables or Figures) should be located in section 3 (materials and methods) since they do not contribute to the overall discussion, besides being described as incidental data.
(2) Authors do not explore statistical methods in order to weigh the effect of geographical origin and plant part on the natural occurrence of specific/different molecules. For example, what do you mean with “obvious differences” in chemical composition from the different position (Line 84)”? at least compounds 48 and 50 (all), 79 (all but seeds) and 53 (all but peel) are evenly distributed.
(3) You stated having identified 85 withanolides (Table 1), yet peaks 86 and 90 were identified in seeds (OF-ESC+) but were not included in Table 1 nor discussed properly.
(4) Results are just described but not discussed properly. Authors should take a look to other supporting reports including yours (DOI): 1002/hlca.200790159, 10.1007/s12272-001-1274-6, https://pdfs.semanticscholar.org/b4b8/1ab33 7 051 ae897ef9136c557eb24d5c2d350.pdf.
Response 7: Done. Thank you for your professional insights.
(1) we have changed the sections 2.1, 2.3, and 2.4 and related Tables or Figures to sections 3.1, 3.3 and 3.5 in our manuscript. (marked red in lines 163-169, 218-224 and 226-230)
(2) I am sorry for some misleading in our manuscript, and we have made corrections in the manuscript. Based on your comments, we have explored the differences of chemical composition in different parts. The results show that during the BPCs of six parts, only seven withanolides compounds (20, 22, 24, 47, 48, 50 and 80) existed in various parts, seventy-eight remaining compounds are unevenly distributed in six locations. What’s more, the compound numbers of flower (42) are twice as seed (21). we have made changes to the relevant statements in our manuscript. (marked red in lines 69, 91-101)
(3) Thanks for your correction, we have made changes in our supporting information. (Figure S1)
(4) Thanks very much for your precious comments, and we have further discussed to the relevant results in our manuscript. (marked red in lines 91-101, 112-115, 127-132 and 137-160)
Point 8: Methods. Please change the following:
(1) Section 3.1. How many plant samples (in fact they should be termed as accessions) correspond to each geographical location?
(2) All equipment (and manufacturers) should be properly annotated (Heating circumfluencer, centrifuge, rotary evaporator, etc.).
(3) Section 3.2. Include structure annotations (lines 173-180) as figure footnote and eliminate from body text.
Response 8: Done. Thanks for your advice.
(1) We have added the number of plant samples from each geographical location in our manuscript. (marked red in lines 170-171)
(2) we have made changes to the relevant equipment in our manuscript. (marked red in lines 184-187)
(3) In our manuscript, we have eliminated structure annotations (section 3.2) from body text and adjusted the numbers of 22 withanolides in Figure 4, which are the same as those in Table 1. (Figure 4)
Point 9: Conclusions You stated, “This was the first qualitative and quantitative analysis of the withanolides in metel L.” – It is not (references 2, 3, 12, 14, 15, 17, 19, 20, 21).
Response 9: Done. Thanks very much for your patience. I am sorry for some misleading, and we have adjusted this sentence as “This is a breakthrough report about simultaneous qualitative and quantitative analysis of 22 withanolides in D. metel L..” in our manuscript. (marked red in lines 22-23 and 251-252)
References:
- Chen, P.H.; Zheng, F.C.; et al. Ethnobotanical study of medicinal plants on arthritis used by Chaoshan in Guangdong. Med Chem (Los Angeles) 2016, 6, 715-723.
- Yang, B.Y.; Guo, R.; et al. New anti-inflammatory withanolides from the leaves of Datura metel L. Steroids 2014, 87, 26-34.
- Guo, R.; Liu, Y.; et al. Withanolides from the leaves of Datura metel L. Phytochemistry 2018, 155, 136-146.
- Ma, L.; Xie, C.M.; et al. Daturametelins H, I, and J: three new withanolide glycosides from Datura metel L. Chem Biodivers 2006, 3, 180-186.
- Alali, F.Q.; Amrine, C.S.M.; et al. Bioactive withanolides from Withania obtusifolia. Phytochem Lett 2014, 9, 96-101.
- Xu, S.; Liu, Y.; et al. Metabolites identification of bioactive compounds daturataturin A, daturametelin I, n-trans-feruloyltyramine, and cannabisin F from the seeds of Datura metel in rats. Front Pharmacol 2018, 9, 731.
Reviewer 3 Report
The Manuscript describes the withanolides content on 6 parts of the medicinal plant Datura metel L., determined by LCMSMS analyses, from 60 specimens collected in 10 different areas of China. Several extraction techniques are compared. Compounds have been identified by comparison with standards purified in by the same group, and quantification of compounds in each part of the plants have been stablished by interpolation in curves of the standards.
In general, experimental design is well designed and executed. Materials and methods are also well described. Results seem correct discussed and concluded.
In general legends of the figures are not in the right place. Referee suggest combining all traces from figure 1 into two figures, in portrait, in two pages: 1A for ESC+ 1B for ESC-, with the 5 parts of the plant in each figure.
There is text in landscape (lines 108-114 and 116-124) should be placed in portrait.
Figure 3 and Supplementary info do have the error bars whereas Figure 4 does not. Please add error-bars to Figure 4.
Regarding the conclusions. All samples have been collected during same time. A seasonal study may indicate different composition in the different parts of the medicinal plant. It is important to differentiate the composition of each medicinal plants along the year. Conclusions may differ for different seasons. ¿Are these medicinal plants harvested during the same season? Authors should add comments on that to the discussion or consider it for the conclusions.
English must be improved a lot! Correct punctuation is needed. Specially the word ‘position’ as it is, is not correctly used in the context. Please change the word ‘position’ to ‘parts of the plant’ and use ‘areas’ instead of ‘parts’ for different geographical origins. Reviewer suggests changing the title into: "… from six parts of the medicinal plant Datura metel L.". The use of the term 'position/s', ‘excavate’ and ‘substitutability’, in the text is misleading and should be reviewed by a native English speaker.
Apart from all that, this reviewer recommends the publication of the work after these minor checks.
Author Response
Point 1: In general legends of the figures are not in the right place. Referee suggest combining all traces from figure 1 into two figures, in portrait, in two pages: 1A for ESC+ 1B for ESC-, with the 5 parts of the plant in each figure.
Response 1: Done. Thanks for your advice. Combined your comments with those of other reviewers, we have adjusted Figure 1 and transferred it in supplementary material. (Figure S1)
Point 2: There is text in landscape (marked red in lines 108-114 and 116-124) should be placed in portrait.
Response 2: Done. Thanks for your advice, we have transferred text direction in our manuscript. (marked red in lines 68-87)
Point 3: Figure 3 and Supplementary info do have the error bars whereas Figure 4 does not. Please add error-bars to Figure 4.
Response 3: Done. Thanks for your advice, we have added error-bars to Figure 4. in our manuscript. (Figure 3)
Point 4: Regarding the conclusions. All samples have been collected during same time. A seasonal study may indicate different composition in the different parts of the medicinal plant. It is important to differentiate the composition of each medicinal plants along the year. Conclusions may differ for different seasons. Are these medicinal plants harvested during the same season? Authors should add comments on that to the discussion or consider it for the conclusions.
Response 4: Done. Thanks for your constructive and pertinent advice. In our research, we harvest the different parts based on the optimal harvesting period of its botany in order to obtain more comprehensive ingredients. The flowers are measured during its flourishing florescence, and leaves, stems and root are collected in vegetative growth stage. The peel and seed are harvested in fruit maturity. Some literatures also support this decision [1,2]. Certainly, as you said, seasonal differences also influence the chemical composition in plants. This will be the topic of our next research.
We have added the corresponding harvest time to Table S3. (marked red in Table S3) We are also added comments on that to the discussion in our manuscript. (marked red in lines154-157)
Point 5: English must be improved a lot! Correct punctuation is needed. Specially the word ‘position’ as it is, is not correctly used in the context. Please change the word ‘position’ to ‘parts of the plant’ and use ‘areas’ instead of ‘parts’ for different geographical origins. Reviewer suggests changing the title into: "… from six parts of the medicinal plant Datura metel L.". The use of the term 'position/s', ‘excavate’ and ‘substitutability’, in the text is misleading and should be reviewed by a native English speaker.
Response 5: Done. Thanks for your correction, our manuscript has been reviewed by a native English speaker to improve the gramma/wording style. (marked red in manuscript)
References:
- Chinese Pharmacopoeia Commission. Chinese pharmacopoeia, 2015 ed.; China Medical Science Press: Beijing, China, 2015; pp. 267.
- Nanjing University of Chinese Medicine. Great Dictionary of Chinese Medicine, 2nd ed.; Shanghai Scientific & Technical Publishers: Shanghai, China, 2006; pp. 1719.
Round 2
Reviewer 2 Report
Thanks for having addressed most of my observations. However, there remain certain aspects in you manuscript that needs to be addressed before publication:
- The manuscript's English grammar and phrasing remain poor. Past/present tenses are misused throughout the text and here are two examples:
- Current (Lines 13 to 15): An UPLC-MS/MS method are applied for identification and content analysis of withanolides in six parts (flower, leaf, stem, root, seed and peel) of Datura metel L. (D metel L.) from ten producing areas in China. Should be: Withanolides from six parts (flower, leaf, stem, root, seed, and peel) of Datura metel L. obtained from ten producing areas in China were identified and quantified by UPLC-MS/MS.
- Current (Lines 94 to 96): The contents of compound 80 are observed variety widely according to the production area, with concentrations that varied from 1670.8 (Dujiangyan) to 10537.7 ng/g (Baotou) (Table S1). Should be: The amount of compound 80 (Daturametelin A) was influenced by the production area, ranging from 1,671 (Dujiangyan) to 10,538 (Baotou) ng/g (Table S1)
Since you already sent your manuscript to an independent colleague, the use of MDPI English editing service is strongly suggested (https://www.mdpi.com/authors/english
- References are not formatted according to Molecules. The ACS style should be followed. Here some examples:
- Kuang, H.-X.; Yang, B.-Y.; Xia, Y.-G.; Wang, Q.-H. Two New Withanolide Lactones from Flos Daturae. Molecules2011, 16, 5833-5839;
- Fan, Y.; Mao, Y.; Cao, S.; Xia, G.; Zhang, Q.; Zhang, H.; Qiu, F.; Kang, N. S5, a Withanolide Isolated from Physalis Pubescens L., Induces G2/M Cell Cycle Arrest via the EGFR/P38 Pathway in Human Melanoma A375 Cells. Molecules2018, 23, 3175.
- Thank you for having included Figure 2 (Heat-Map); however, this should be discussed in depth within the text. For example, according to Euclidean distances, there are clearly more pronounced differences by region than by plant part which, by the way, is accentuated from the top (leaf, flower) to the bottom (root) of the plant.
- There is no information in materials and methods (Section 3) on the statistical (e.g. ANOVA) and chemometric (e.g. Hierarchical clustering, heat-map).
Author Response
Point 1: The manuscript's English grammar and phrasing remain poor. Past/present tenses are misused throughout the text and here are two examples:
Current (Lines 13 to 15): An UPLC-MS/MS method are applied for identification and content analysis of withanolides in six parts (flower, leaf, stem, root, seed and peel) of Datura metel L. (D metel L.) from ten producing areas in China. Should be: Withanolides from six parts (flower, leaf, stem, root, seed, and peel) of Datura metel L. obtained from ten producing areas in China were identified and quantified by UPLC-MS/MS.
Current (Lines 94 to 96): The contents of compound 80 are observed variety widely according to the production area, with concentrations that varied from 1670.8 (Dujiangyan) to 10537.7 ng/g (Baotou) (Table S1). Should be: The amount of compound 80 (Daturametelin A) was influenced by the production area, ranging from 1,671 (Dujiangyan) to 10,538 (Baotou) ng/g (Table S1)
Since you already sent your manuscript to an independent colleague, the use of MDPI English editing service is strongly suggested (https://www.mdpi.com/authors/english
Response 1: Done. Thanks for your correction, and we have revised our manuscript by your advice. (marked red in lines 13-15 and 96-97) Additionally, our manuscript has been submitted to the company you suggested for language editing. (marked red in manuscript) The invoice has been uploaded to the submission system. (Figure 1)
Point 2: References are not formatted according to Molecules. The ACS style should be followed.
Kuang, H.-X.; Yang, B.-Y.; Xia, Y.-G.; Wang, Q.-H. Two New Withanolide Lactones from Flos Daturae. Molecules 2011, 16, 5833-5839;
Fan, Y.; Mao, Y.; Cao, S.; Xia, G.; Zhang, Q.; Zhang, H.; Qiu, F.; Kang, N. S5, a Withanolide Isolated from Physalis Pubescens L., Induces G2/M Cell Cycle Arrest via the EGFR/P38 Pathway in Human Melanoma A375 Cells. Molecules 2018, 23, 3175.
Response 2: Done. Thank you very much for your valuable comments, and we have adjusted reference style in our manuscript. (marked red in Reference).
Point 3: Thank you for having included Figure 2 (Heat-Map); however, this should be discussed in depth within the text. For example, according to Euclidean distances, there are clearly more pronounced differences by region than by plant part which, by the way, is accentuated from the top (leaf, flower) to the bottom (root) of the plant.
Response 3: Done. Thank you for your professional insights. Certainly, as you said, for the total withanolide content of each producing area, the difference of area is more pronounced than plant part. According to cluster analysis, they are also divided into three groups in Figure 2. S1 (Fuyang), S2 (Haikou) and S7 (Zhaotong) make up one group. S3 (Xingtai), S4 (Dujiangyan) and S5 (Harbin) are another group, and the final group is S6 (Ganzhou), S8 (Baotou), S9 (Baoji) and S10 (Jinhua). Dujiangyan (S4,`x = 81789 ng/g), Jinhua (S10,`x = 67177 ng/g) and Fuyang (S1,`x = 41607 ng/g) are three areas with the top, second, and bottom total withanolide amounts, and they are also evenly distributed in three branch groups. This further proves that the origin is more obvious than the parts. We have added the relevant discussion in our manuscript. (marked red in lines 122-130)
Point 4: There is no information in materials and methods (Section 3) on the statistical (e.g. ANOVA) and chemometric (e.g. Hierarchical clustering, heat-map).
Response 4: Done. Thank you for your professional insights. Group differences are determined by one-way analysis of variance (ANOVA) using GraphPad Prism 7 (GraphPad, U.S.A.) and P<0.05 is considered statistically significant. Hierarchical clustering and correlation heat-map were constructed using R language package at the OmicsShare platform (https://www.omicshare.com/tools/). And we have adjusted relevant discussion and information in our manuscript. (marked red in lines 130-133, 255-258, and Figure S2)
